# Knowledge, attitude, and preventive practices towards COVID-19 and associated factors among adult hospital visitors in South Gondar Zone Hospitals, Northwest Ethiopia

Zebader Walle Belete[1☯], Gete Berihun[2☯]*, Awoke Keleb[2], Ayechew Ademas[2],
Leykun Berhanu[2], Masresha Abebe[2], Adinew Gizeyatu[2], Seada Hassen[2],
Daniel Teshome[3], Mistir Lingerew[2], Alelgne Feleke[2], Tarikuwa Natnael[2],
Metadel Adane[2]

1 Department of Public Health, College of Health Sciences, Debre Tabor University, Debre Tabor, Ethiopia,
2 Department of Environmental Health, College of Medicine and Health Sciences, Wollo University, Dessie,
Ethiopia, 3 Department of Anatomy, College of Medicine and Health Sciences, Wollo University, Dessie,
Ethiopia

☯ These authors contributed equally to this work.
* geteberihun@gmail.com

org/10.1371/journal.pone.0250145

Infectious Diseases Lazzaro Spallanzani-IRCCS,
ITALY

## Abstract

### Background

Coronavirus disease 2019 (COVID-19) is currently the critical health problem of the globe,
including Ethiopia. Visitors of healthcare facilities are the high-risk groups due to the pres-
ence of suspected and confirmed cases of COVID-19 in the healthcare setting. Increasing
the knowledge, attitude, and practices towards COVID-19 prevention among hospital visi-
tors are very important to prevent transmissions of the pandemic despite the lack of evi-
dence remains a challenge in Ethiopia. Therefore, this study was designed to investigate
the status of knowledge, attitude, and preventive practice towards COVID-19 and associ-
ated factors among hospital visitors in South Gondar Zone Hospitals, Northwest Ethiopia.

### Methods

A facility-based cross-sectional study design was employed during August 1 to 30, 2020
from randomly selected 404 adult hospital visitors in South Gondar Zone Hospitals, North-
west Ethiopia. The data was collected using interviewer-administered questionnaire. The
outcome of this study was good or poor knowledge, positive or negative attitude and good or
poor preventive practice towards COVID-19. Three different binary logistic regression mod-
els with 95% CI (Confidence interval) was used for data analysis. For each mode, bivariable
analysis (crude odds ratio [COR]) and multivariable analysis (adjusted odds ratio [AOR])
was used during data analysis. From the bivariable analysis, variables with a *p*-value <0.25
were retained into the multivariable logistic regression analysis. From the multivariable logis-
tic regression analysis, variables with a significance level of *p*-value <0.05 were taken as

**Data Availability Statement:** All relevant data are within the paper and its Supporting information files.

**Funding:** The author(s) received no specific funding for this work.

**Competing interests:** The authors have declared that no competing interests exist.

**Abbreviations:** $ USD, United States Dollars; AOR, Adjusted odds ratio; CI, Confidence interval; COR, Crude odds ratio; COVID-19, Coronavirus disease 2019.

factors independently associated with knowledge, attitude and preventive practices towards COVID-19.

## Main findings

About 69.3% of the respondents had good knowledge, 62.6% had a positive attitude, and 49.3% had good preventive practice towards the prevention of COVID-19. We found that factors significantly associated with good knowledge about COVID-19 were educational status who can read and write (AOR = 2.78; 95%CI: 1.18–6.56) and college and above (AOR = 6.15; 95%CI: 2.18–17.40), and use of social media (AOR = 2.96; 95%CI: 1.46–6.01). Furthermore, factors significantly associated with a positive attitude towards COVID-19 includes the presence of chronic illnesses (AOR = 5.00; 95%CI: 1.71–14.67), training on COVID-19 (AOR = 3.91; 95%CI: 1.96–7.70), and peer/family as a source of information (AOR = 2.45; 95%CI: 1.06–5.63). Being a student (AOR = 7.70; 95%CI: 1.15–15.86) and participants who had a good knowledge on COVID-19 (AOR = 4.49; 95%CI: 2.41–8.39) were factors significantly associated with good practice towards COVID-19.

## Conclusion

We found that knowledge, attitude, and preventive practices towards prevention of COVID-19 among adult hospital visitors were low. Therefore, we recommended that different intervention strategies for knowledge, attitude and preventive practices are urgently needed to control the transmission of COVID-19 among adult hospital visitors. Health education of those who could not read and write about COVID-19 knowledge issues and advocating use of social media that transmit messages about COVID-19 are highly encouraged to increase the good knowledge status of adult hospital visitors. Furthermore, providing training about COVID-19 prevention methods and using various sources of information about COVID-19 will help for improving positive attitude towards COVID-19 prevention, whereas for increasing the status of good preventive practices towards COVID-19, improving the good knowledge about COVID-19 of adult hospital visitors are essential.

## Introduction

Corona virus 2019 (COVID-19) is a rapidly emerging pandemic respiratory disease caused by a novel Coronavirus of Severe Acute Respiratory Syndrome (SARS/COV-2). The disease was reported initially in Wuhan city, Hubei Province, China at the end of December 2019 [1–3]. Later on, World Health Organization (WHO) announces the disease as a public health emergency of international concern at the end of January 2020 and then declared as a global pandemic on March 11 [4–6]. Two days later, the government of Ethiopia reported the first confirmed case of COVID-19 [7, 8].

COVID-19 transmits mainly through droplets, airborne transmission, and contact between humans [6, 9–11]. The major sign and symptoms of COVID-19 cases are fever, dry cough, fatigue, myalgia, shortness of breath, and dyspnoea [4–6]. The Severe cases of the disease may lead to the developments of cardiac injury, respiratory failure, acute respiratory distress syndrome, and death. Elders and patients with chronic medical illnesses like hypertension, cardiac

disease, lung disease, cancer, or diabetes have been identified as potential risk factors for disease severity and mortality [6–11].

According to the Worldometer report, as of October 6, 2020, 9:54 am, COVID-19 spreads to more than 214 countries across the world. Worldwide, a total of 35,707,844 confirmed cases were reported. Of them, 26,907,997 were recovered and 1,049,700 died of the pandemic [12]. In the case of Ethiopia, 79,437 confirmed cases of COVID-19 were reported. Of this, 1,230 and 34,016 died and recovered, respectively [13].

Due to the absence of cure [6–11] prevention is recommended as the only strategy to prevent the spread of COVID-19. Different COVID-19 prevention measures are implemented such as respiratory hygiene, hand washing, social distancing, use of Personal Protective Equipment, and environmental disinfection [6, 14–17]. WHO designed different guidelines and online training sessions to increase the awareness of the community towards the prevention of the pandemic [18]. But still, information was deficient mainly for vulnerable groups [6, 17].

The government of Ethiopia has also implemented different prevention measures. Later on, the country declares and enforces a state of emergency for about six months since March 2020. But most populations of Ethiopia perceived that as the disease was eliminated since the termination of a state of emergency. Therefore, prevention measures towards COVID-19 are becoming neglected from time to time. Healthcare facilities are one of the vulnerable areas for transmission of COVID-19. As a result, visitors of healthcare facilities are one of the victim groups of population for COVID-19 due to close contact with suspected and confirmed cases of the disease. Occupation, sex, age, family size, marital status, residence, average monthly income, and alcohol consumption were some of the factors affecting knowledge, attitude and practices of COVID-19 [19, 20].

Even though many researchers were conducted on knowledge, attitude and preventive practices towards prevention of COVID-19, there is limited evidence on knowledge, attitude and preventive practices of healthcare facility visitors. Therefore, assessing knowledge, attitude, and practice are important measures for identifying gaps and taking intervention accordingly [4]. Therefore, the study was designed to assess the knowledge, attitude, and preventive practice of adult visitors towards COVID-19 prevention in South Gondar Zone Hospitals, Northwest Ethiopia.

## Methods and materials

### Study area description

South Gondar zone is one of the 13 administrative zones in the Amhara regional state of Ethiopia. Debre Tabor is its capital town which is located at 597 km and 105 km from Addis Ababa and Bahir Dar, respectively. According to the Central Statistical Agency projection of 2014–2017, the total population of the study area was 2,484,929 in 2017 of which 1,257,323 (50.6%) were males while 1,227,606 (49.4%) were females [21].

In South Gondar Zone, there are one comprehensive referral hospital found in Debre Tabor Town and seven district government hospitals found in Addis Zemen, Mekane Eyesus, Andabet, Ebenat, Arb Gebeya, Nefas Mewcha, and Wegeda twons. In addition, there are 98 government health centers, and 76 private clinics in South Gondar Zone [22]. According to the reports of hospitals in the South Gondar Zone, the average number of monthly visitors during COVID-19 was 13,440.

### Study design and population

Institution-based cross-sectional study design was conducted among hospital visitors in South Gondar Zone Hospitals, Northwest Ethiopia from August 1 to 30, 2020. The source population

was all adult visitors with age of 18 and above in hospitals of South Gondar zone while the study population was adult visitors of Debre Tabor comprehensive referral hospital and Mekane Eyesus hospitals. Those who were seriously ill at the time of data collection were not included in the study.

## Sample size determination and sampling methods

The sample size was determined using the single population proportion formula by taking the following assumptions.

$$n = \frac{(z_{a/2})^2 * p(1-p)}{d^2}$$

$Z_{\alpha/2}$ is the standard normal variable value at (1-α)% confidence level (α is 0.05 with 95% CI, $Z_{\alpha/2}$ = 1.96), an estimate of the proportion of knowledge, attitude and preventive practice, was considered as 50% as there were no similar studies conducted and 5% margin of error was considered. The sample size became 384 and after considering 10% non-response rates, the adequate sample final sample size becomes 422.

After selecting the two hospitals randomly out of the 8 hospitals, we proportionally allocated the sample size based on the total estimated visitors of hospitals in the last three months. Then, 303 sample size was allocated for Debre Tabor comprehensive referral hospital and 117 for Mekane Eyesus hospitals. Then hospital visitors data during the previous 3 months in the emergency ward, surgical ward, medical ward, gynecology/obstetrics ward, and pediatrics ward considered for sample size allocation for each hospital's departments. Finally, the randomly selection of visitors for each ward was applied until the allocated sample size was achieved.

## Operational definitions

**Good or poor knowledge.** Knowledge was measured by using 15 items of questions consisted of signs and symptoms, risk groups and prognosis, method of transmission, and /preventive methods towards COVID-19. Each question was consisted of 'Yes', 'No', and 'I do not know' options. Respondents who answered correctly were given 1 point while others were given 0 points. The total knowledge score ranges from 0–15 and a cut-off level of ≥12 (80% and above) was considered as good knowledge while <12 (80%) was considered as poor knowledge towards COVID-19 prevention [23].

**Positive or negative attitude.** Attitude was measured by using 11 items of questions about precautions methods for preventive practices towards COVID-19 and the response was categorized based on 3 scale measurements with agree (3 points), neutral (2 points), and disagree (1 point). The score of attitude varied from 11 to 33, with an overall mean score of ≥ 27 (81.8%) was considered as a positive attitude about precautions for preventive practices towards COVID-19, whereas a score of less than 27 (81.8%) was considered as negative attitude towards COVID-19 prevention [27].

**Good or poor preventive practice.** The preventive practice was measured using 10 items of questions and those who respond as yes were given 1 point while no was marked as 0. The total prevention practice score ranges from 0–10 and a score with a cut-off ≥ 8 (80%) was considered as good practice while <8 was taken as a poor practice [24, 25].

## Data collection and quality assurance

Data were collected using a pre-tested structured questionnaire which was adapted from published articles in reputable journals and WHO COVID-19 guidelines [26–31]. The questionnaire consists of five sections including; part I: socio-demographic characteristics of the participants; part II; Pre-existing medical condition and sources of information towards COVID-19; part III: knowledge of the participants; part IV: Attitude of the participants; and part V: Prevention practice of COVID-19. The tool was prepared in the English version and translated to the Amharic version (local language), and re-translated back to English to ensure consistency. The tool was pre-tested using 5% of the final sample size in Andabet hospital visitors to establish the validity of the questionnaire. Based on the pre-test, appropriate amendments such as order arrangement of questions, editing of unclear questions, and avoiding irrelevant questions were done accordingly.

The data was collected using interviewer-administered method using four BSc in Environmental Health professionals and supervised by two Public Health experts (one supervisor for one hospital). Two days of training were given for data collectors and supervisors on the overall aim of the study, contents of the tool, data collection procedures and about ethical issues. Supervision was carried out on daily basis, and appropriate corrections of the collected data were done accordingly. Furthermore, double data entry was done to control data entry errors and data cleaning was carried before statistical analysis. The reliability coefficient of Cronbach's alpha was 0.76 which is an acceptable range.

## Statistical analysis

Data was entered into EpiData version 4.6 and exported to the Statistical Package of the Social Science (SPSS) version 25.0 for data cleaning analysis. Descriptive statistics such as frequencies and percentages were calculated for categorical variables and mean with standard deviations for continuous variables to examine the overall distribution.

Associations between independent variables with the outcomes of knowledge, attitudes, and preventive practices towards COVID-19 were determined using a binary logistic regression model at 95% CI (Confidence interval) independently. We used three different logistic regression models: The first model (Model 1) identified factors associated with good knowledge about COVID-19, the second model (Model II) identified factors associated with positive attitudes towards precautions measures of COVID-19 and the third model (Model III) identified factors associated factors with good preventive practices towards COVID-19. For each model, bivariable analysis with (crude odds ratio [COR]) and multivariable analysis (adjusted odds ratio [AOR]) was used.

From the bivariable analysis, variables with a $p$-value $<0.25$ were retained into the multivariable logistic regression analysis. From the multivariable analysis of each model, variables with a significance level of $p$-value $<0.05$ were taken as factors independently associated with knowledge, attitude, and practices towards COVID-19. The presence of multicollinearity among independent variables was checked using standard error at the cutoff value of 2 and we found that a maximum standard error of 0.97, which indicated no multi-collinearity. Model fitness was checked using the Hosmer-Lemeshow test for Models I, II and III knowledge, had a $p$-value of 0.650, 0.871, and 0.913, respectively, and indicated that all models were fit.

## Ethics approval and consent to participate

The study was approved by the ethical review committee of College of Health Sciences, Debre Tabor University. Permission to conduct the study was obtained from the respective hospital managers of the study site. Before the data collection, the purpose of the study was explained

and written informed consent was obtained from the study participants. Individuals who were volunteer to participate in the study were also told as they have the right to withdraw from the study at any stage of the interview. The confidentiality of the study participants was ensured by avoiding possible identifiers. Data collectors wear a facemask and keep a physical distancing of two feet. Facemask was provided for the study participants who did not wear it during the data collection.

## Result

### Socio-demographic characteristics of hospital visitors

A total of 404 visitors participated in the study with a response rate of 95.7%. Nearly one-third 117 (29.0%) of the study participants were lived in rural areas. Nearly two-thirds 241 (59.7%) of the study participants were females and about one-fifth 92 (22.8%) of the hospital visitors were in the age range of 20–29 years. Furthermore, the educational status of 66 (16.3%) of the study participants cannot read and write, 68 (16.8%) of the study participants occupation were farmers. and 117 (29.0%) of the respondents. Fifty six (13.9%) of the participants had either one or more chronic medical illness history (Table 1).

### Knowledge of hospital visitors towards COVID-19 prevention

More than two-thirds 280 (69.3%; 95%CI; 65.1–73.8%) of the visitors had good knowledge, whereas 124 (30.7%; 95%CI; 27.2–34.9%) of them had poor knowledge about COVID-19. Almost all 388 (96.0%) of the participants heard about COVID-19 and more than three-fourth 322 (79.7%) of the participants knew as COVID-19 is a viral disease and 339 (83.9%) of them knew the major sign and symptoms of COVID-19 cases. Furthermore, more than three-fourth 320 (79.2%) of the participants knew that elders, those who had a chronic medical illness and being obese are more likely to have severe cases of COVID-19. Similarly, 283 (70.0%) of the respondents knew that COVID-19 can be transmitted from one person to another even in the absence of COVID-19 (Table 2).

### Attitude of hospital visitors towards COVID-19 prevention

About two-thirds 62.6% (95%CI; 57.2–67.6%) of the hospital visitors had a positive attitude towards COVID-19 prevention, whereas 37.4% (95%CI: 32.4–42.8%) respondents had negative attitude towards COVID-19 prevention. About half 203 (50.3%) of the participants agree that the black race is not protective against COVID-19. Similarly, less than half 180 (44.6%) of the participants agreed that Ethiopia is in a good position to contain the spread of the COVID-19. About two-thirds, 274 (67.8%) of the participants believed COVID-19 does not cause stigma. More than half 221 (54.7%) of the respondents agree that they can get infected with COVID-19 if they contacted infected patients despite their good immunity. On the other hand, 55 (13.6%) of the respondents believed that COVID-19 has occurred as a result of our sin (Table 3).

### Preventive practice of hospital visitors towards COVID-19 prevention

Half of the respondents 199 (49.3%) practiced the recommended COVID-19 prevention methods. The majority 378 (93.6%) of the participants washed their hands with water and soap for at least 20 seconds. Furthermore, almost nine out of ten respondents avoid handshaking practice for the prevention of COVID-19. But a relatively lower number of 338 (83.7%) participants used facemasks when they leave their home and 333 (82.4%) practiced respiratory hygiene

**Table 1. Socio-demographic characteristics of adult visitors in hospitals of South Gondar zone Northwestern Ethiopia, August 1 to 30, 2020.**

| Variable | Category | Frequency (*n*) | Percent (%) |
|---|---|---|---|
| **Sex** | Male | 163 | 40.3 |
| | Female | 241 | 59.7 |
| **Age (years)** | <20 | 24 | 5.9 |
| | 20–29 | 92 | 22.8 |
| | 30–39 | 111 | 27.5 |
| | 40–49 | 94 | 23.2 |
| | 50–59 | 48 | 11.9 |
| | ≥60 | 35 | 8.7 |
| **Religion** | Muslim | 30 | 7.4 |
| | Orthodox | 331 | 82.0 |
| | Protestant | 43 | 10.6 |
| **Marital status** | Single | 83 | 20.5 |
| | Married | 295 | 73.0 |
| | Divorced | 26 | 6.5 |
| **Educational status** | Cannot read and write | 66 | 16.3 |
| | Read and write | 95 | 23.5 |
| | Primary (1–8 grade) | 36 | 8.9 |
| | Secondary (9–12 grade) | 29 | 7.2 |
| | College and above | 178 | 44.1 |
| **Occupation** | Farmer | 68 | 16.8 |
| | Student | 45 | 11.1 |
| | Unemployed | 55 | 13.6 |
| | Government employer | 129 | 32.0 |
| | Private business worker | 107 | 26.5 |
| **Resident** | Urban | 287 | 71.0 |
| | Rural | 117 | 29.0 |
| **Monthly income ($, USD)** | <13.82 | 127 | 31.4 |
| | 13.82–55.26 | 101 | 25.0 |
| | >55.26 | 176 | 43.6, |
| **History of chronic medical illness** | Yes | 56 | 13.9 |
| | No | 348 | 86.1 |
| **Obtained training on COVID 19** | Yes | 137 | 33.9 |
| | No | 267 | 66.1 |
| **Use social media** | Yes | 252 | 62.4 |
| | No | 152 | 37.6 |
| **Peer as a source of information of COVID19** | Yes | 345 | 85.4 |
| | No | 59 | 14.6 |
| **Use TV/radio as a source of information** | Yes | 321 | 79.5 |
| | No | 83 | 20.5 |
| **Use religious institution as source of information** | Yes | 130 | 32.2 |
| | No | 274 | 67.8 |

*1 $USD (United States Dollars) exchange rate was 36.1914 Ethiopian Birr (ETB) during August 1 to 30, 2020.

while coughing and sneezing. Furthermore, less than half 177(43.8%) of the participants applied to keep the recommended physical distance for the prevention of COVID-19. Staying at home was also another challenge and only less than one-third 121(30%) of the participants applied it (Table 4).

**Table 2. Knowledge of hospital visitors towards COVID-19 prevention in hospitals of South Gondar zone, Northwestern Ethiopia, August 1 to 30, 2020.**

| Questions* | Response | | | | | |
|---|---|---|---|---|---|---|
| | Yes | | No | | I do not know | |
| | Frequency (*n*) | Percent (%) | Frequency (*n*) | Percent (%) | Frequency (*n*) | Percent (%) |
| Did you hear about COVID-19? | 388 | 96.0 | 16 | 4.0 | 0 | 0 |
| COVID-19 is a viral disease. | 322 | 79.7 | 40 | 9.9 | 42 | 10.4 |
| The major sign and symptoms of COVID-19 are dry cough, fever, and shortness of breathing. | 339 | 83.9 | 33 | 8.2 | 32 | 7.9 |
| Runny nose and sneezing are less common symptoms of COVID-19. | 275 | 68.1 | 88 | 21.8 | 41 | 10.1 |
| Elder, those who have a chronic medical illness and obese are more likely to sever the case of COVID- 19. | 320 | 79.2 | 57 | 14.1 | 27 | 6.7 |
| Currently, there is no effective cure for COVID-19. | 331 | 81.9 | 50 | 12.4 | 23 | 5.7 |
| COVID-19 virus can spread via respiratory droplets. | 375 | 92.8 | 29 | 7.2 | | |
| Eating and contacting wild animals would result COVID-19 infection | 308 | 76.2 | 53 | 13.2 | 43 | 10.6 |
| Persons with COVID 19 virus can transmit the virus to others when a fever is not present | 283 | 70.0 | 71 | 17.6 | 50 | 12.4 |
| Proper washing hand with soap and water is one method of preventing COVID-19. | 375 | 92.8 | 18 | 4.5 | 11 | 2.7 |
| Wearing general masks can prevent one from acquiring infection by the COVID 19 virus | 354 | 87.6 | 35 | 8.7 | 15 | 3.7 |
| Children and young adults must take measures to prevent the infection by Covid 19 virus | 337 | 83.4 | 45 | 11.1 | 22 | 5.5 |
| To prevent the infection by COVID 19 virus individuals should avoid going to crowded places such as bus parks and avoid public transportation | 352 | 87.1 | 50 | 12.4 | 2 | .5 |
| People who have contact with someone infected with COVID 19 virus should be immediately isolated in a proper place in general the observation period is 14 days | 273 | 67.6 | 95 | 23.5 | 36 | 8.9 |
| Isolation and treatment of people who are infected with the COVID 19 virus are effective ways to reduce the spread of the virus | 295 | 73.0 | 80 | 19.8 | 29 | 7.2 |

*Mean± standard deviation = 12.25±2.45; Minimum = 2 and maximum = 15

**Table 3. Attitude of adult visitors towards COVID-19 prevention in hospitals of South Gondar zone, Northwest Ethiopia, August 1 to 30, 2020.**

| Questions* | Agree | | Neutral | | Disagree | |
|---|---|---|---|---|---|---|
| | Frequency (*n*) | Percent (%) | Frequency (*n*) | Percent (%) | Frequency (*n*) | Percent (%) |
| Black races are not protected from COVID 19 disease. | 203 | 50.3 | 146 | 36.1 | 55 | 13.6 |
| Wearing a well-fitting face mask are effective in preventing COVID 19 virus | 268 | 66.4 | 81 | 20.0 | 55 | 13.6 |
| Hand wash can prevent you from COVID 19 virus | 321 | 79.4 | 77 | 19.1 | 6 | 1.5 |
| Ethiopia is in a good position to contain COVID 19 virus | 180 | 44.6 | 144 | 35.6 | 80 | 19.8 |
| COVID 19 is not stigma and I should not hide my infection | 274 | 67.8 | 90 | 22.3 | 40 | 9.9 |
| If I get infected with COVID 19, I will go to the hospital as advised. | 221 | 54.7 | 141 | 34.9 | 42 | 10.4 |
| I can get infected with COVID 19 if I contacted an infected patient despite my good immunity. | 230 | 56.9 | 100 | 24.8 | 74 | 18.3 |
| COVID 19 is fatal | 215 | 53.2 | 105 | 26.0 | 84 | 20.8 |
| During the outbreak of COVID 19 eating well cooked and safely handled meat is safe. | 249 | 61.6 | 96 | 23.8 | 59 | 14.6 |
| COVID 19 patients should share their recent travel history with a health care provider. | 256 | 63.4 | 85 | 21.0 | 63 | 15.6 |
| Do you think that the cause of Covid-19 is not spiritual/ is it happened because of our sin? | 262 | 64.9 | 87 | 21.5 | 55 | 13.6 |

* Mean ±standard deviation = **27.11±4.08**; Minimum = **17**; Maximum = **33**

**Table 4. Preventive practice of adult visitors towards COVID-19 prevention in hospitals of South Gondar zone, Northwest Ethiopia, August 1 to 30, 2020.**

| Questions* | Yes | | No | |
|---|---|---|---|---|
| | Frequency (*n*) | Percent (%) | Frequency (*n*) | Percent (%) |
| Do you avoid handshaking to prevent covid 19? | 363 | 89.9 | 41 | 10.1 |
| Have you washed your hands often with soap and water for at least 20 seconds especially after you have been in a public place or after blowing your nose, coughing, or sneezing? | 378 | 93.6 | 26 | 6.4 |
| If soap and water are not readily available, are you applying a hand sanitizer that contains at least 60% alcohol? | 309 | 76.5 | 95 | 23.5 |
| Do you wear face masks repeatedly when you leave your home? | 338 | 83.7 | 66 | 16.3 |
| Do you coughing and sneezing into the elbow or within clothing? | 333 | 82.4 | 71 | 17.6 |
| In recent days have you avoid going to any crowded place? | 281 | 69.6 | 123 | 30.4 |
| Do you avoid eating raw animal products to prevent the COVID 19 virus? | 336 | 83.2 | 68 | 16.8 |
| Do you avoid touching your mouth nose and eyes with unwashed hands? | 323 | 80.0 | 81 | 20.0 |
| Do you keep your self 2m away from the others when you got to the public area? | 177 | 43.8 | 227 | 56.2 |
| Do you stay at your home after the emergent of covid 19? | 121 | 30.0 | 283 | 70.0 |

*Mean ±standard deviation was 7.32±1.60 for the correctly responded questions; minimum questions correctly answered was 1 and maximum questions correctly answered was 10

## Factors associated with knowledge, attitude, and preventive practice towards COVID-19 from multivariable analysis

A multi-variable analysis from the first model indicated that educational status and use of social media as a source of information were statistically significant with the knowledge of COVID-19. The fining revealed that those who can read and write were 2.78 times more likely to have good knowledge on COVID-19 prevention methods than those who could not read and write. Similarly, participants who have college and above educational level were 6.15 (AOR = 6.15; 95%CI: 2.18–17.40) times more likely to have good knowledge than those who could not read and write. Furthermore, participants who used social media as a source of information towards COVID-19 were 2.96 (AOR = 2.96; 95%CI: 1.46–6.01) times more likely to have good knowledge than those who did not use social media (Table 5).

A multi-variable analysis from the second model revealed that those who had primary education were 6.49 (AOR = 6.49; 95%CI: 1.52–27.78) times more likely to have a positive attitude than those who could not read and write while being college and above graduated were 6.91 (AOR = 6.91; 95%CI: 2.58–14.5) times more likely to have a positive attitude than the corresponding reference group. Furthermore, visitors who had chronic medical illnesses were 5 times (AOR = 5; 95%CI: 1.71–14.67) more likely to have a positive attitude than those who did not have a chronic illness. Furthermore, participants who took training on COVID-19 prevention were 3.9 (AOR = 3.9; 95%CI: 1.96–7.70) times more likely to have a positive attitude than those who didn't take the training. Additionally, participants who used peer as a source of information towards COVID-19 prevention were 2.45 (AOR = 2.45; 1.06–5.63) times more likely to have a positive attitude than those who didn't use peers as a source of information for COVID-19 prevention (Table 6).

From the multivariable analysis of model three, we found that being a student was 7.7 times (AOR = 7.7; 95%CI: 1.15–51.86) more likely to have a good practice than farmers. Furthermore, participants who had good knowledge were 4.49 (AOR = 4.49; 95%CI: 2.41–8.39) times more likely to have a good practice about COVID-19 prevention than those who poor knowledge (Table 7).

**Table 5. Factors associated with knowledge towards COVID-19 prevention among adult visitors in hospitals of South Gondar zone, Northwest Ethiopia, August 1 to 30, 2020.**

| Variable | Knowledge status | | COR (95% CI) | AOR (95% CI) | P-value |
|---|---|---|---|---|---|
| | Good | Poor | | | |
| **Age (years)** | | | | | |
| <20 | 12 | 12 | 1 | 1 | |
| 20–29 | 70 | 22 | 3.18(1.25–8.09) | 1.98(0.54–7.29) | 0.312 |
| 30–39 | 82 | 29 | 2.83(1.14–6.99) | 1.08(0.30–3.87) | 0.091 |
| 40–49 | 58 | 39 | 1.61(0.65–3.97) | 1.06(0.31–3.68) | 0.921 |
| 50–59 | 33 | 15 | 2.20(0.80–6.02) | 1.21(0.30–4.82) | 0.793 |
| ≥60 | 25 | 10 | 2.50(0.84–7.40) | 0.69(0.16–2.95) | 0.610 |
| **Marital status** | | | | | |
| Single | 61 | 22 | 1 | 1 | |
| Married | 205 | 90 | 0.82(0.48–1.42) | 0.88(0.42–1.82) | 0.724 |
| Divorced | 14 | 12 | 0.42(0.17–1.05) | 0.92(0.29–2.98) | 0.891 |
| **Education** | | | | | |
| Cannot read and write | 25 | 41 | 1 | 1 | |
| Read and write | 54 | 41 | 2.16(1.14–4.12) | 2.78(1.18–6.56) | 0.021 |
| Primary (1–8 grade) | 26 | 10 | 4.26(1.76–10.31) | 2.42(0.56–10.44) | 0.245 |
| Secondary (9–12 grade) | 21 | 8 | 4.31(1.66–11.18) | 1.54(0.25–9.56) | 0.656 |
| College and above | 154 | 24 | 10.52(10.52–5.45) | 6.15(2.18–17.40) | 0.001 |
| **Occupation** | | | | | |
| Farmer | 23 | 45 | 1 | 1 | |
| Student | 33 | 12 | 5.38(2.35–12.34) | 1.64(0.28–9.72) | 0.591 |
| Currently unemployed | 38 | 17 | 4.37(2.04–9.36) | 1.50(0.49–4.58) | 0.488 |
| Government worker | 111 | 18 | 12.07(5.95–24.48) | 0.83(0.16–4.19) | 0.827 |
| Private business | 75 | 32 | 4.59(2.39–8.80) | 0.91(0.25–3.30) | 0.896 |
| **Resident** | | | | | |
| Urban | 216 | 71 | 2.52(1.60–3.96) | 1.43(0.75–2.71) | 0.281 |
| Rural | 64 | 53 | 1 | 1 | |
| **Monthly income ($, USD)** | | | | | |
| <13.82 | 73 | 54 | 1 | 1 | |
| 13.82–55.26 | 67 | 34 | 1.46(0.85–2.51) | 1.272(0.52–3.09) | 0.600 |
| >55.26 | 140 | 36 | 2.88(1.73–4.78) | 1.29(0.46–3.60) | 0.630 |
| **Obtained training on COVID 19** | | | | | |
| Yes | 113 | 24 | 2.82(1.70–4.67) | 1.74(0.89–3.42) | 0.110 |
| No | 167 | 100 | 1 | 1 | |
| **Use social media** | | | | | |
| Yes | 204 | 48 | 4.25(2.72–6.65) | 2.96(1.46–6.01) | 0.003 |
| No | 76 | 76 | 1 | 1 | |
| **Use of peer as a source of information of COVID19** | | | | | |
| Yes | 252 | 93 | 3.00(1.71–5.27) | 1.09(0.48–2.51) | 0.840 |
| No | 28 | 31 | 1 | 1 | |
| **Use TV/radio as a source of information** | | | | | |
| Yes | 243 | 78 | 3.87(2.34–6.40) | 1.07(0.43–2.65) | 0.885 |
| No | 37 | 46 | 1 | 1 | |
| **Use religious institution as source of information** | | | | | |
| Yes | 98 | 32 | 1.55(0.97–2.48) | 0.93(0.50–1.73) | 0.834 |
| No | 182 | 92 | 1 | 1 | |

1, reference category

**Table 6. Factors associated with attitude towards COVID-19 prevention among adult visitors in hospitals of South Gondar zone, Northwest Ethiopia, in August 1 to 30, 2020.**

| Variable | Attitude | | COR (95% CI) | AOR (95% CI) | p-value |
|---|---|---|---|---|---|
| | Positive | Negative | | | |
| **Age (years)** | | | | | |
| <20 | 12 | 12 | 1 | 1 | |
| 20–29 | 57 | 35 | 1.63(0.66–4.02) | 0.42(0.12–1.46) | 0.171 |
| 30–39 | 66 | 45 | 1.47(0.61–3.56) | 0.51(0.14–1.82) | 0.302 |
| 40–49 | 57 | 37 | 1.54(0.63–3.79) | 0.94(0.28–3.21) | 0.933 |
| 50–59 | 32 | 16 | 2.00(0.74–5.44) | 1.10(0.28–4.27) | 0.895 |
| ≥60 | 29 | 6 | 4.83(1.47–15.87) | 1.65(0.33–8.42) | 0.554 |
| **Religion** | | | | | |
| Muslim | 25 | 5 | 3.27(1.05–10.20) | 2.18(0.50–9.58) | 0.302 |
| Orthodox | 202 | 129 | 1.02(0.53–1.96) | 1.49(0.64–3.48) | 0.361 |
| Protestant | 26 | 17 | 1 | 1 | |
| **Education** | | | | | |
| Cannot read and write | 22 | 44 | 1 | 1 | |
| Read and write | 47 | 48 | 1.96(1.02–3.76) | 2.39(0.99–5.79) | 0.053 |
| Primary (1–8 grade) | 27 | 9 | 6.00(2.41–14.93) | 6.49(1.52–27.78) | 0.012 |
| Secondary (9–12 grade) | 21 | 8 | 5.25(2.01–13.74) | 2.32(0.39–13.74) | 0.35 |
| College and above | 136 | 42 | 6.48(3.49–12.01) | 6.91(2.58–14.50) | <0.001 |
| **Occupation** | | | | | |
| Farmer | 22 | 46 | 1 | 1 | |
| Student | 36 | 9 | 8.36(3.44–20.36) | 1.87(0.33–10.72) | 0.481 |
| Currently unemployed | 33 | 22 | 3.14(1.50–6.58) | 0.54(0.18–1.68) | 0.292 |
| Government worker | 105 | 24 | 9.15(4.66–17.96) | 0.61(0.12–3.05) | 0.553 |
| Private business | 57 | 50 | 2.38(1.26–4.50) | 0.29(0.07–1.12) | 0.075 |
| **Resident** | | | | | |
| Urban | 193 | 94 | 1.95(1.26–3.02) | 1.23(0.66–2.23) | 0.514 |
| Rural | 60 | 57 | 1 | 1 | |
| **Monthly income ($, USD)** | | | | | |
| <13.82 | 72 | 55 | 1 | 1 | |
| 13.82–55.26 | 61 | 40 | 1.17(0.69–1.98) | 0.89(0.33–2.38) | 0.826 |
| >55.26 | 120 | 56 | 1.64(1.02–2.63) | 0.57(0.19–1.70) | 0.312 |
| **History of chronic illness** | | | | | |
| Yes | 48 | 8 | 4.19(1.92–9.12) | 5.00(1.71–14.67) | 0.003 |
| No | 205 | 143 | 1 | 1 | |
| **Obtained training on COVID-19** | | | | | |
| Yes | 113 | 24 | 4.27(2.59–7.05) | 3.9(1.96–7.70) | <0.001 |
| No | 140 | 127 | 1 | 1 | |
| **Use social media** | | | | | |
| Yes | 179 | 73 | 2.59(1.70–3.93) | 1.20(0.59–2.44) | 0.631 |
| No | 74 | 78 | 1 | 1 | |
| **Use of peer as a source of information about COVID-19** | | | | | |
| Yes | 231 | 114 | 3.41(1.92–60.5) | 2.45(1.06–5.63) | 0.042 |
| No | 22 | 37 | 1 | 1 | |
| **Use TV/radio as a source of information about COVID-19** | | | | | |
| Yes | 221 | 100 | 3.52(2.13–5.81) | 2.091(0.85–5.16) | 0.113 |
| No | 32 | 51 | 1 | 1 | |

(*Continued*)

**Table 6.** (Continued)

| Variable | Attitude | | COR (95% CI) | AOR (95% CI) | *p*-value |
|---|---|---|---|---|---|
| | Positive | Negative | | | |
| **Use religious institution as source of information about COVID-19** | | | | | |
| Yes | 100 | 30 | 2.64(1.64–4.23) | 1.725(0.93–3.21) | 0.094 |
| No | 153 | 121 | 1 | 1 | |

1, reference category

## Discussion

We conducted institution based cross-sectional study to examine the status of knowledge, attitude, preventive practices and associated factors among hospital visitors in South Gondar Zone Hospitals. We found that 69.3% of the study participates had good knowledge, 62.6% of them had a positive attitude and less than half (49.3%) of had good preventive practice towards the prevention of COVID-19.

The finding of this study revealed that 69.3% (CI; 65.1–73.8) of the participants had good knowledge on COVID-19 prevention which was in line with the study conducted in India (70.0%) [32]. On the other hand, this study finding was lower than a multicenter study conducted among health care workers in Ethiopia with 88.2% [15] and Nigerian residents in an urban setting (99.7%) [36]. This deviation may be due to variations in socio-demographic characteristics of the study population and sources of information towards COVID-19.

In this finding, about 81.67% of the knowledge questions were correctly replied to by the respondents. This finding was in line with the study conducted in Saudi Arabia (80.5%) (4) and in Nigeria (77.36) [33]. The finding of this study was lower than the study conducted in China (90%) [34]. This discrepancy may be due to variation in the study population's characteristics, government commitment, and health care system. On the contrary, this study result was higher than in the Egyptian population (71.26%) [35]. This discrepancy might be due to the variation in socio-demographic characteristics of the population.

This study also revealed that about 80% of participants knew that the elderly, those who had chronic medical illnesses, and obese are more likely to develop severe cases of COVID-19. This finding was slightly higher than the study conducted in Ethiopia (72.5%) [11]. This variation may be due to the change in the study period, socio-demographic characteristics of the study population, and coverage of awareness creation towards COVID-19 prevention. Even though children and young adults are vulnerable groups, only 83.4% of the participants knew that these groups need to take preventive measures towards COVID-19. Neglecting such types of population may wide-spreading the transmission of the pandemic [11].

Regarding the attitudes, 62.6% (95% CI; 57.2–67.6) of respondents had a positive attitude towards COVID-19 prevention which was lower than the study conducted in Ethiopia (94.7%) [15], Nigeria 79.5% [36], and Pakistan (82.16%) [37]. This discrepancy may be due to a change in the socio-demographic characteristics of the study population, government commitment towards COVID-19. On the other hand, less than half (44.6%) of the participants believed that the government of Ethiopia can control the spread of COVID-19 within a short time. This finding was lower than the study conducted in China 97.1% [23] and India 87.2% [38]. This deviation may be due to the variation in the quality of the health care system, socio-demographic characteristics of the study population, and government preparedness towards the control of the COVID-19 pandemic.

**Table 7. Factors associated with preventive practice towards COVID-19 prevention among adult visitors in hospitals of South Gondar zone, Northwest Ethiopia, in August 1 to 30, 2020.**

| Variable | Preventive practice status | | COR (95% CI) | AOR (95% CI) | P-value |
|---|---|---|---|---|---|
| | Good | Poor | | | |
| **Age (years)** | | | | | |
| <20 | 9 | 15 | 1 | 1 | |
| 20–29 | 48 | 44 | 1.82(0.72–4.57) | 0.87(0.24–3.38) | 0.842 |
| 30–39 | 49 | 62 | 1.32(0.53–3.26) | 0.43(0.12–1.67) | 0.221 |
| 40–49 | 41 | 53 | 1.29(0.51–3.24) | 0.75(0.20–2.56) | 0.671 |
| 50–59 | 27 | 21 | 2.14(0.79–5.85) | 0.99(0.23–4.24) | 0.982 |
| ≥60 | 25 | 10 | 4.17(1.38–12.58) | 1.31(0.28–6.11) | 0.731 |
| **Education** | | | | | |
| Cannot read and write | 17 | 49 | 1 | 1 | |
| Read and write | 25 | 70 | 1.03(0.50–2.11) | 0.93(0.36–2.43) | 0.881 |
| Primary (1–8 grade) | 16 | 20 | 2.31(0.98–5.44) | 0.39(0.08–1.80) | 0.231 |
| Secondary (9–12 grade) | 17 | 12 | 4.08(1.62–10.27) | 0.82(0.16–4.19) | 0.821 |
| College and above | 124 | 54 | 6.62(3.50–12.52) | 1.90(0.67–5.17) | 0.212 |
| **Occupation** | | | | | |
| Farmer | 6 | 62 | 1 | 1 | |
| Student | 24 | 21 | 11.81(4.25–32.83) | 7.70(1.15–15.86) | 0.042 |
| Currently unemployed | 20 | 35 | 5.91(2.17–16.08) | 2.35(0.58–9.57) | 0.234 |
| Government worker | 93 | 36 | 26.70(10.62–67.12) | 2.49(0.42–14.61) | 0.316 |
| Private business | 56 | 51 | 11.35(4.52–28.47) | 2.15(0.45–10.2) | 0.348 |
| **Resident** | | | | | |
| Urban | 166 | 121 | 3.49(2.19–5.56) | 1.54(0.79–3.00) | 0.217 |
| Rural | 33 | 84 | 1 | 1 | |
| **Monthly income ($, USD)** | | | | | |
| <13.82 | 39 | 88 | 1 | 1 | |
| 13.82–55.26 | 50 | 51 | 2.21(1.29–3.81) | 2.05(0.71–5.93) | 0.196 |
| >55.26 | 110 | 66 | 3.76(2.32–6.12) | 1.99(0.62–6.39) | 0.253 |
| **Obtained training on COVID-19** | | | | | |
| Yes | 86 | 51 | 2.30(1.51–3.51) | 0.88(0.47–1.64) | 0.684 |
| No | 113 | 154 | 1 | 1 | |
| **Use social media** | | | | | |
| Yes | 160 | 92 | 5.04(3.23–7.87) | 1.54(0.76–3.10) | 0.231 |
| No | 39 | 113 | 1 | 1 | |
| **Use of peer as a source of information about COVID19** | | | | | |
| Yes | 184 | 161 | 3.35(1.80–6.25) | 0.78(0.31–1.97) | 0.613 |
| No | 15 | 44 | 1 | 1 | |
| **Use of Tv/radio as source of information about COVID-19** | | | | | |
| Yes | 185 | 136 | 6.70(3.62–12.41) | 1.45(0.53–3.96) | 0.462 |
| No | 14 | 69 | 1 | 1 | |
| **Knowledge** | | | | | |
| Poor knowledge | 22 | 102 | 1 | 1 | |
| Good knowledge | 177 | 103 | 7.97(4.73–13.41) | 4.49(2.41–8.39) | <0.001 |
| **attitude** | | | | | |
| Negative attitude | 44 | 107 | 1 | 1 | |
| Positive attitude | 155 | 98 | 3.85(2.50–5.93) | 1.04(0.58–1.86) | 0.068 |

1, reference category

According to the WHO report, the government of Ethiopia scored 52% towards the COVID-19 preparedness response [39] which supports the finding of this study. Furthermore, this study also indicated that almost two-thirds of the respondents believed that the pandemic of COVID-19 leads to the development of social stigma which was lower than a study conducted in Ethiopia at 77% [15] and 83.8% [11]. This deviation may be due to differences in socio-demographic characteristics of the study population and study period. On the contrary, this study finding was higher than the study conducted in the Peruvian population 59.1% [40]. This variation may be due to a change in the socio-demographic characteristics of the study population, study period, awareness creation towards COVID-19, and the burden of the pandemic. The social stigma may be developed due to fear of its mortality and high communicability. The history of social stigma due to pandemics was not a new phenomenon [41, 42].

Regarding the prevention practice of COVID-19, the overall practice score of the respondents was 73.2% which was higher than the study conducted in Ethiopia [25]. The finding of this study revealed that only half of the respondents 49.3% had a positive preventive practice of COVID-19. The finding of this study was lower than other studies conducted in Ethiopia [15, 25] and China [43]. This variation may be due to the change in the study setting, socio-demographic characteristics of the study population, and occupation of the study participant (being a health professional vs. general population), and the commitment of the government towards the prevention of COVID-19. Furthermore, most of the participants 93.6% washed their hands with water and soap for at least 20 seconds which was in line with a study conducted in Nigeria 96.4% [36]. On the contrary, this finding was lower than a study conducted in Nigeria 87.9% [32]. This deviation may be to a change in access and utilization of handwashing facilities in health care facilities.

Furthermore, 83.7% of the participants used face masks for the prevention of COVID-19 which were consistent with the study conducted in Nigeria 84.4% [32], and 82.3% [36]. This finding also revealed that less than half (43.8%) of the respondents applied the recommended physical distance of 2 meters when they go to public crowded areas. This finding was lower than the study conducted in Nigeria 83% [32] and 92.7% [36]. This variation may be due to a change in the socio-demographic characteristic of the study population, the burden of the disease, awareness of the community towards the COVID-19 pandemic, and population way of life.

The finding also revealed that more than two-thirds 70% of the respondents avoid going to crowded places after the emergence of COVID-19 which was higher than the finding in Nigeria 58.9% [32]. in addition to this, 82.4% of the respondents practiced respiratory hygiene which was lower than the finding in India (97.7%) [44]. The variation might be due to a change in a study setting, heterogeneity of population perception of the community, knowledge towards COVID-19, and burden of confirmed COVID-19 cases. Above all, the most common problem which was not applied by the participants was staying at home and only less than one-third (30%) after the occurrence of COVID-19. The possible justification for this might be due to the uncontrolled of the pandemic for a long period of time, poor preparedness of the community towards the pandemic, and subsistence way of life of the community.

## Conclusion

Generally, knowledge, attitude, and practice towards the prevention of COVID-19 was low. Factors significantly associated with good knowledge on COVID-19 were educational status who can read and write and college and above and use of social media. Factors significantly associated with a positive attitude towards COVID-19 include presence of chronic illnesses, training on COVID-19, and peer/family as a source of information. Being a student and

participants who had a good knowledge on COVID-19 were factors significantly associated with good practice towards COVID-19. Hence, intervention strategies that could improve the knowledge, attitude, and practice status towards COVID-19 preventions are urgently needed to control the transmission of COVID-19. Health education for improving the status of knowledge, attitude, and preventive practice towards COVID-19 should be a priority agenda for hospitals, and health bureau administrators and other concerned stakeholders. Furthermore, providing training about COVID-19 prevention methods and using various sources of information about COVID-19 will help for improving positive attitude towards COVID-19 prevention, whereas for increasing the status of good preventive practices towards COVID-19, improving the good knowledge of adult hospital visitors are essential. Therefore, hospital administrators should work in collaboration with other concerned stakeholders to enhance the knowledge, attitude, and practice of hospital visitors towards COVID-19 prevention.

## Supporting information

**S1 Data.**
(DOCX)

**S2 Data.**
(DOCX)

**S3 Data.**
(XLSX)

## Acknowledgments

Our heartfelt gratitude extends to Debre Tabor University for providing ethical clearance for this study. Debra Tabor comprehensive referral hospital and Mebane Eyesus hospital administrators are duly acknowledged for their permission to conduct this study and for the information we received when needed. We also thank the study participants who gave valuable information and their kind cooperation during this study. We are also grateful to data collectors and supervisors for their commitment during the data collection.

## Author Contributions

**Conceptualization:** Zebader Walle Belete, Gete Berihun, Awoke Keleb, Ayechew Ademas, Metadel Adane.

**Data curation:** Zebader Walle Belete, Gete Berihun, Awoke Keleb, Ayechew Ademas, Metadel Adane.

**Formal analysis:** Zebader Walle Belete, Gete Berihun, Awoke Keleb, Ayechew Ademas, Metadel Adane.

**Funding acquisition:** Gete Berihun.

**Investigation:** Zebader Walle Belete, Gete Berihun, Awoke Keleb, Ayechew Ademas, Leykun Berhanu, Masresha Abebe, Adinew Gizeyatu, Seada Hassen, Daniel Teshome, Mistir Lingerew, Alelgne Feleke, Tarikuwa Natnael, Metadel Adane.

**Methodology:** Zebader Walle Belete, Gete Berihun, Awoke Keleb, Ayechew Ademas, Leykun Berhanu, Masresha Abebe, Adinew Gizeyatu, Seada Hassen, Daniel Teshome, Mistir Lingerew, Alelgne Feleke, Tarikuwa Natnael, Metadel Adane.

**Project administration:** Zebader Walle Belete, Gete Berihun, Awoke Keleb, Ayechew Ademas, Leykun Berhanu, Masresha Abebe, Adinew Gizeyatu, Seada Hassen, Daniel Teshome, Mistir Lingerew, Alelgne Feleke, Tarikuwa Natnael, Metadel Adane.

**Resources:** Zebader Walle Belete, Gete Berihun, Awoke Keleb, Ayechew Ademas, Leykun Berhanu, Masresha Abebe, Adinew Gizeyatu, Seada Hassen, Daniel Teshome, Mistir Lingerew, Alelgne Feleke, Tarikuwa Natnael, Metadel Adane.

**Software:** Zebader Walle Belete, Gete Berihun, Awoke Keleb, Ayechew Ademas, Leykun Berhanu, Masresha Abebe, Adinew Gizeyatu, Seada Hassen, Daniel Teshome, Mistir Lingerew, Alelgne Feleke, Tarikuwa Natnael, Metadel Adane.

**Supervision:** Zebader Walle Belete, Gete Berihun, Awoke Keleb, Ayechew Ademas, Leykun Berhanu, Masresha Abebe, Adinew Gizeyatu, Seada Hassen, Daniel Teshome, Mistir Lingerew, Alelgne Feleke, Tarikuwa Natnael, Metadel Adane.

**Validation:** Zebader Walle Belete, Gete Berihun, Awoke Keleb, Ayechew Ademas, Leykun Berhanu, Masresha Abebe, Adinew Gizeyatu, Seada Hassen, Daniel Teshome, Mistir Lingerew, Alelgne Feleke, Tarikuwa Natnael, Metadel Adane.

**Visualization:** Zebader Walle Belete, Gete Berihun, Awoke Keleb, Ayechew Ademas, Leykun Berhanu, Masresha Abebe, Adinew Gizeyatu, Seada Hassen, Daniel Teshome, Mistir Lingerew, Alelgne Feleke, Tarikuwa Natnael, Metadel Adane.

**Writing – original draft:** Zebader Walle Belete, Gete Berihun, Awoke Keleb, Ayechew Ademas, Metadel Adane.

**Writing – review & editing:** Zebader Walle Belete, Gete Berihun, Ayechew Ademas, Metadel Adane.

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
