## [Decision Letter · Decision Letter 0]

27 Jan 2021

PONE-D-20-36453

Preventive Practice and Associated Factors towards COVID-19 among Medical Visitors in Hospitals of South Gondar Zone, Northwest Ethiopia

PLOS ONE

Dear Dr. Berihun,

Thank you for submitting your manuscript to PLOS ONE. After careful consideration, we feel that it has merit but does not fully meet PLOS ONE’s publication criteria as it currently stands. Therefore, we invite you to submit a revised version of the manuscript that addresses the points raised during the review process.

We look forward to receiving your revised manuscript.

Kind regards,

Francesco Di Gennaro

Academic Editor

PLOS ONE

Additional Editor Comments:

dear authors follow reviewer suggestion to improve your paper

Journal Requirements:

6. We note you have included a table to which you do not refer in the text of your manuscript. Please ensure that you refer to Table 2 in your text; if accepted, production will need this reference to link the reader to the Table.

Reviewers' comments:

Reviewer's Responses to Questions

**Comments to the Author**

1. Is the manuscript technically sound, and do the data support the conclusions?

Reviewer #1: Partly

Reviewer #2: Yes

Reviewer #3: Yes

2. Has the statistical analysis been performed appropriately and rigorously? 

Reviewer #1: Yes

Reviewer #2: Yes

Reviewer #3: Yes

3. Have the authors made all data underlying the findings in their manuscript fully available?

Reviewer #1: Yes

Reviewer #2: Yes

Reviewer #3: Yes

4. Is the manuscript presented in an intelligible fashion and written in standard English?

Reviewer #1: Yes

Reviewer #2: No

Reviewer #3: Yes

5. Review Comments to the Author

Reviewer #1: Dear Editor,

I carefully read the article by Barium et al., which is interesting a quite well done.

My remarks are the following:

General comment

• What is the difference between patient and visitors in your study?

• The document has not page number please incorporate

• Some sentence are incomplete which need intensive editing

Abstract section

1. Objective

….please add space between ‘of’ and ‘south’ as ‘To assess COVID-19 preventive practice and associated factors among visitors in 30 hospitals of South Gondar Zone, Northwest Ethiopia.’ It is also page 2 line 42 please add space.

2. Method

… It say “The questionnaire was pre-tested in 5% of the final sample size to establish the validity of the data collection instrument. The data were collected using face-to-face interviews by considering physical distancing and wearing of face masks. The data was entered in Epi-data version 3.1 and exported to Statistical Package for Social Science (SPSS) Version 25 for analysis.” It is better delete and replace by tool of outcome variable measurement.

‘Bi-variate Crude Odd Ration (COR) with 95% confidence intervals (CI) and p-values of less than 0.25 were applied to select candidate variables for multi-variable analysis. Then, multi-variable Adjusted Odd Ratio (AOR) using binary logistic regression analysis at a p-value of less than 0.05 at 95% CI was….’shall replace this paragraph by this ‘Logistic regression was applied to assess the association between dependent and explanatory variables. The association was interpreted using the adjusted odds ratio (AOR) and 95% confidence interval (CI)’.

3. Result

‘Multi-variable (AOR) logistic

…….. (AOR=2.96; 95% CI: 1.46, 6.01) were significantly associated with knowledge of COVID-1.’ This is incomplete and does not give sense. It is better rewrite as ‘……were associated significantly with visitors’ knowledge about prevention of COVID-19.’ The same is true for altitude.

Line 53-54 says…… good knowledge on COVID-19 (AOR=4.49; 54 95% CI: 2.41-8.39) were significantly associated. Does not make sense please add subject and verb like participant who had good knowledge ….

4. Conclusion

The conclusion did not in line the finding. Please conclude according to the finding.

Methods and Materials section

1. Line 34-37 ‘The patient flow data were estimated by reviewing the patients' logbook in the last three months and the average number of the patient for a month was calculated to determine the interval. Then, we used a systematic random sampling technique to select study participants of the study’. This paragraph is not clear. How to reach to apply systematic random sampling technique? What is the sampling frame? Is your study population are patients or patients attendance or any visitor of the hospital?

2. Under outcome and explanatory variables: please delete sensitive words like poor knowledge, attitude, and practice. It shall be replaced with ‘favorable/unfavorable’

3. Delete subtopic of ‘operational definition.’ This is already state in the outcome variables. Please avoid bolding words like ‘good knowledge, poor knowledge, positive attitude, and Practice.’

4. Line 65 and 66 ‘A pre-test was conducted using 5% (21) of the

final sample size in the Andabet district to establish the validity of the questionnaire and amendment was made accordingly.’ What type of amendment you made? Can you explain that amendment?

5. Under Statistical analysis line 77-78: what is the different between bi- variate and multi-variable? What do you mean ‘multi-variable’?

Result section

1. Use one of result presentation. Almost all tables are explained in the text. Please follow rule of text and table presentation together.

2. Line 208 & 209: ‘Similarly, 283 (70%) of the respondents knew that COVID-19 can be transmitted from one person to another even in the absence of COVID-19 (Table 1).’ Table 1 presents socio-demographic characteristics of the participants but not knowledge of participants. Please cite the table appropriately.

3. Line 234-238: ‘The finding of the study revealed seven out of ten 280 (69.3%) respondents had good knowledge towards COVID-19 while 253 (62.6%) had a positive attitude towards COVID-19. The finding of the study showed the pillar of prevention practice was much lower and only half 199(49.3%) of the participants had a score of good prevention practice of COVID-19 (Figure 1).’ This paragraph is not clear. This is better rewrite as’...280 (69.3% of study participants had favorable knowledge towards COVID-19 prevention. ‘... almost half of the study participants 119 (49.3%) are practiced the recommended COVID-19 prevention methods.’

4. Line 242-247: this paragraph is not clear. The sentence are not provide full information. For example ‘The finding revealed that those who can read and write were 2.78 times more likely to have good knowledge than those who can‘t read and write.’ To more clear rewrite as “The finding

revealed that those who can read and write were 2.78 times more likely to have good knowledge towards COVID-19 prevention methods than those who can‘t read and write.” The same is true for others and the next paragraph also.

5. Line 273 &274: ‘This discrepancy 274 might be due to Spatio-temporal variation.’ This is not a justifiable reason to the knowledge people towards novel coronal virus discrepancy of between Ethiopia and Egyptian population. Please search another justification of this discrepancy.

6. Line 279 & 280: This deviation may be due to the change in the study population (health care professionals vs. general population) and residents of the study population. This make confuse reader please rephrase again

Reviewer #2: The outcome variable and the title is not congruent. If your outcome variables are knowledge, attitude and practice towards COVID -19 prevention, your title should be modified to KAP of visitors towards COVID-19 prevention. otherwise, if your title is focused only practice and associated factor, you have to include knowledge and attitude as part of associated factor, rather than the outcome variable.

Reviewer #3: Result and discussion part 1. Please use software modeling for clear elaboration the topic of CVID 19

Examples OLS Model, add other better software

2. Adding Images as possible

3. Edit grammatical error and other

6. PLOS authors have the option to publish the peer review history of their article (what does this mean?). If published, this will include your full peer review and any attached files.

Reviewer #1: No

Reviewer #2: **Yes: **You can disclose my full name as a reviewer of this manuscript.

Reviewer #3: No

---

## [Author Response · Author response to Decision Letter 0]

12 Mar 2021

Date: March 12 2021

Manuscript ID: PONE-D-20-36453R1 

Preventive Practice and Associated Factors towards COVID-19 among Medical Visitors in Hospitals of South Gondar Zone, Northwest Ethiopia

Corresponding authors: Gete Berihun (MSc)

Dear Dr, Francesco Di Gennaro (PhD) 

Academic Editor

PLOS ONE

Thank you for your letter dated January 27, 2021 with a decision of major revision needed. We were pleased to know that our manuscript was considered potentially acceptable for publication in PLoS ONE, subject to adequate revision as requested by the reviewers, academic editors and the journals. Based on the instructions provided in your letter, we uploaded the file of the rebuttal letter; the marked up copy of the revised manuscript highlighting the changes made in the original submitted version and the clean copy of the revised manuscript. 

We have revised the manuscript by modifying the abstract, introduction, methods, results, discussion and other sections, based on the comments made by the reviewers and using the journal guidelines. Therefore, we have marked in red color all the changes made during the revision process. Appended to this letter is our point-by-point response (rebuttal letter) to the comments made by the reviewers. 

We have agreed with almost all the comments and questions raised by the reviewers and academic editors. We also provided justification for those comments and questions for disagreeing. We would like to take this opportunity to express our thanks to the reviewers for their valuable comments and to thank you for allowing us to resubmit a revision of the manuscript. 

I hope that the revised manuscript is accepted for publication in PLoS ONE. 

Sincerely yours,

Gete Berihun (Wollo University) 

Response to the Journal Requirements Questions

Response to editor 

Question #1 Please ensure that your manuscript meets PLOS ONE's style requirements, including those for file naming.

Response: Thank you for this remark. We re-formatted the revised manuscript using the PLoS ONE format guidelines. The whole content of the manuscript, including the abstract, introduction, methods, discussion and reference are formatted using the guidelines (Please see the revised version for each section).

Question #2. Data availability

Response. We have attached the data on the supplementary information 

Question #3. Please amend either the title on the online submission form (via Edit Submission) or the title in the manuscript so that they are identical. 

Response: Thank you for your comment. We made the title identical on the online submission and the title in the manuscript. 

Question #4. Please amend either the abstract on the online submission form (via Edit Submission) or the abstract in the manuscript so that they are identical.

Response: Thank you for your comment and we did it accordingly. 

Question #5. Your ethics statement should only appear in the Methods section of your manuscript. If your ethics statement is written in any section besides the Methods, please move it to the Methods section and delete it from any other section. Please ensure that your ethics statement is included in your manuscript, as the ethics statement entered into the online submission form will not be published alongside your manuscript.

Response: Thank you. We did it. 

Question #6. We note that you have included a table to which you do not refer in the text of your manuscript. Please ensure that you refer to Table 2 in your text; if accepted, production will need this reference to link the reader to the Table.

Response: We have accepted the comments and hence the text written in Table 1 is changed in to Table 2. 

Response to reviewer 1

Question #1. What is the difference between patient and visitors in your study?

Response: Sorry for the confusion about the two terms. We understand that we used the two terms interchangeably and now we updated the manuscript by hospital visitors. the patient is all persons who came to health care facilities for medical treatment. But visitors are any persons who came to the health care facilities for different purposes including seeking of medical treatment. Therefore, to avoid confusion we used visitors since the study was done on visitors, not only to patients (see the revised version). 

Question #2. The document has not page number please incorporate

Response: Based on your comment, we gave the page number accordingly. Thank you. 

Question # 3. Some sentences are incomplete which need intensive editing.

Response: We tried to assess errors like incomplete sentence, grammatical and language error from title up to discussion of the manuscript. As a result, the amendment was done accordingly in the revised version of the manuscript. 

Questions #4. In the abstract section please add space between ‘of’ and ‘south’ as ‘To assess COVID-19 preventive practice and associated factors among visitors in 30 hospitals of South Gondar Zone, Northwest Ethiopia.

Response: Thank you for your comment. We made a correction in the revised manuscript. 

Question #5. In abstract section, “The questionnaire was pre-tested in 5% of the final sample size to establish the validity of the data collection instrument. The data were collected using face-to-face interviews by considering physical distancing and wearing of face masks. The data was entered in Epi-data version 3.1 and exported to Statistical Package for Social Science (SPSS) Version 25 for analysis.” It is better delete and replace by tool of outcome variable measurement.

Response: Based on your comment we removed less important points from the abstract and amendment was done accordingly (see the revised new version of the manuscript).

Question #6. In abstract section, Bi-variate Crude Odd Ration (COR) with 95% confidence intervals (CI) and p-values of less than 0.25 were applied to select candidate variables for multi-variable analysis. Then, multi-variable Adjusted Odd Ratio (AOR) using binary logistic regression analysis at a p-value of less than 0.05 at 95% CI was shall be rephrased.

Response: Thank you for this key comment, we revised accordingly and please see the data analysis in page 9 and 10. 

Question #7. In the abstract section, (AOR=2.96; 95% CI: 1.46, 6.01) were significantly associated with knowledge of COVID-1.’ This is incomplete and does not give sense. It is better rewrite it again.

Response: We have accepted your comment hence this sentence rewritten as …were associated significantly with visitors’ knowledge towards the prevention of COVID-19 (Please see the revised version of the abstract). 

Question #8. In abstract section, the conclusion did not in line the finding. Please conclude according to the finding.

Response: We have accepted your comment. The conclusion was amended based on the finding of the study (see the revised version of the manuscript).

Question #9. In Methods and Materials section, the patient flow data were estimated by reviewing the patients' logbook in the last three months and the average number of the patient for a month was calculated to determine the interval. Then, we used a systematic random sampling technique to select study participants of the study’. This paragraph is not clear. How to reach to apply systematic random sampling technique? What is the sampling frame? Is your study population are patients or patients attendance or any visitor of the hospital?

Response: We found that our way of writing was confusing. Sorry for the mistakes. After selecting the two hospitals randomly out of the 8 hospitals, we proportionally allocated sample size based on total estimated visitors of hospitals in the last three months. Then, 303 sample size was allocated for Debre Tabor general hospitals and 117 for Mekane Eyesus hospitals. Then hospitals visitors flow data during the previous 3 months in emergency ward, surgical ward, medical ward, gynecology/obstetrics ward and pediatrics ward considered for sample size allocation for each hospitals departments. Finally, randomly selection of visitors for each ward was selected until the allocated sample size was achieved (See the revised version in page 7. 

 Question #10. Under outcome and explanatory variables: please delete sensitive words like poor knowledge, attitude, and practice. It shall be replaced with ‘favorable/unfavorable’

Response: We accepted your reflections, however, most studies used good/poor knowledge, positive and negative attitude and good/poor practice. We used these terms accordingly throughout the paper. 

Question #11. Delete subtopic of ‘operational definition.’ This is already state in the outcome variables. Please avoid bolding words like ‘good knowledge, poor knowledge, positive attitude, and Practice

Response: Thank you for your comment; we deleted subtopic of operational definition

Question#12. A pre-test was conducted using 5% (21) of the final sample size in the Andabet district to establish the validity of the questionnaire and amendment was made accordingly.’ What type of amendment you made? Can you explain that amendment?

Response: The pre-test is aimed for amendment of measuring tool. Therefore, some of the amendments were arrangement, editing of unclear questions, and avoiding irrelevant questions. 

Question 13. Under Statistical analysis: what is the different between bi- variate and multi-variable? What do you mean ‘multi-variable’?

Response: Bivariate analysis refers one independent variable with outcome variables. However, multivariable means that more than one independent variables with the outcome variable. From the adjusted analysis, all variables that has a p-value less than 0.25 were included into the adjusted multivariable analysis to control confounders. In our study, in the case of this study we used bivariable and multivariable analysis. Furthermore, the word bivariate analysis was changed to bivariable analysis throughout the manuscript. 

Question#14. Result section, use one of result presentation. Almost all tables are explained in the text. Please follow rule of text and table presentation together.

Response: Based on your comments we reduced more than half of the explanation. As a result, only pertinent finding of the study was explained (Please see all result section). 

Question #15. In result section, similarly, 283 (70%) of the respondents knew that COVID-19 can be transmitted from one person to another even in the absence of COVID-19 (Table 1).’ Table 1 presents socio-demographic characteristics of the participants but not knowledge of participants. Please cite the table appropriately.

Response: We made a correction accordingly (See the revised version of the manuscript). 

Question #16. The finding of the study revealed seven out of ten 280 (69.3%) respondents had good knowledge towards COVID-19 while 253 (62.6%) had a positive attitude towards COVID-19. The finding of the study showed the pillar of prevention practice was much lower and only half 199(49.3%) of the participants had a score of good prevention practice of COVID-19 (Figure 1).’ This paragraph is not clear.

Response: This idea is rewritten as’...280 (69.3% of study participants had favorable knowledge towards COVID-19 prevention. ‘... almost half of the study participants 119 (49.3%) are practiced the recommended COVID-19 prevention methods.’

Question #17. In result section, the finding revealed that those who can read and write were 2.78 times more likely to have good knowledge than those who can‘t read and write is not clear and should be re-write.

Response: We have rewritten as “The finding revealed that those who can read and write were 2.78 times more likely to have good knowledge towards COVID-19 prevention methods than those who can‘t read and write”.

Question #18. In discussion section, this discrepancy 274 might be due to Spatio-temporal variation.’ This is not a justifiable reason to the knowledge people towards novel coronal virus discrepancy of between Ethiopia and Egyptian population. Please search another justification of this discrepancy.

Response: Based on comment we tried to elaborate better justification for variation in this study with the finding of other researches conducted in different parts of the world. 

Question #19. In discussion section, this deviation may be due to the change in the study population (health care professionals vs. general population) and residents of the study population. This makes confuse reader please rephrase again

Response: This idea is rewritten as” the deviation may be due to the difference in the study subjects. In the present study, the study subjects were visitors of Hospitals while the study conducted in Egypt were only health care professionals.

Response to reviewer 2 

 Question #1. The outcome variable and the title is not congruent. If your outcome variables are knowledge, attitude and practice towards COVID -19 prevention, your title should be modified to KAP of visitors towards COVID-19 prevention. Otherwise, if your title is focused only practice and associated factor, you have to include knowledge and attitude as part of associated factor, rather than the outcome variable.

Response: Based on the comment, we tried to make the title in line with its outcome variable. Therefore the title is modified in to KAP of visitors towards COVID-19 prevention while the outcome variables of the study are Knowledge, attitude, and practice towards COVID-19 prevention 

Response to reviewer 3

Question #1. Result and discussion part 1. Please use software modeling for clear elaboration the topic of CVID 19 Examples OLS Model, add other better software

Response: We have already used logistic regression analysis using crude odds ratio (COR) and adjusted odds ratio (AOR) for determining the associated factors with the outcome variables. Associations between independent variables and knowledge, attitudes and practices towards COVID-19 were determined using a binary logistic regression model at 95% CI (Confidence interval). We used three different logistic regression models: The first model (Model 1) identified factors associated with good knowledge about COVID-19, the second model (Model II) identified factors associated with favorable attitudes and the third model (Model III) identified factors associated factors with good preventive practices towards COVID-19. For each model, bivariable analysis with (COR [crude odds ratio]) and multivariable analysis (AOR [adjusted odds ratio]) was used. 

From the bivariable analysis, variables with a p-value <0.25 were retained into the multivariable logistic regression analysis. From the multivariable analysis of each model, variables with a significance level of p-value <0.05 were taken as factors independently associated with knowledge, attitude and practices towards COVID-19. The presence of multicollinearity among independent variables was checked using standard error at the cutoff value of 2 and we found that a maximum standard error of 0.97, which indicated no multi-collinearity. Model fitness was checked using the Hosmer-Lemeshow test for Model I, Model II and Model III and found a p-value of 0.650, 0.871 and 0.913, respectively and indicated that all models were fit.

We hope that the data analysis we used above is very sufficient to our study, which we could able to explain the result and discussion as we did it. 

Question #2. Adding Images as possible

Response: We have 7 Tables and including more Figure is repeating of the result of the Table in another forms. 

Question #3. Edit grammatical error and other

Response: This comment was also raised by other reviewers. We tried to revise of grammar, language, and punctuation errors starting from the title of the manuscript up to discussion (see the revised version of the manuscript). We appreciate your comment. 

Response to reviewer 4 

Question #1. What does medical visitor mean? is that for only visiting the medical ward? if not, it is better to say among visitors.

Response: The study subjects all visitors of the health care facilities. So that, the title is modified to Knowledge, Attitude, and Practices towards COVID-19 and associated factors among hospital Visitors in South Gondar Zone Hospitals, Northwest Ethiopia

\\Question #2. In the background section, please also include information on preventive practices of COVID-19.

Response: The title of the revised is modified to Knowledge, Attitude, and Practices towards COVID-19 and associated factors among hospital Visitors in South Gondar Zone Hospitals, Northwest Ethiopia. Therefore the emphasis is given not only for prevention measures but also for knowledge and attitude towards prevention of COVID-19. But in the revised manuscript, we elaborate detail COVID-19 prevention measures. 

Question #3. Replace the word "face-to-face administered" by "interviewer administered"

Response: The phrase “face-to-face administered” is replaced by “interviewer administered”. (See the revised version of the manuscript).

Question #4. Please include the confidence intervals for these percentages with 95% CI.

Response: Knowledge, attitude, and practice of respondents are presented in the result and discussion section of the revised manuscript. According to this study, 69.3% (95%CI;65.1-73.8%) had a favorable knowledge, 62.6% (95%CI;57.2-67.6) had a favorable attitude, and 49.3%(95%CI;) had a favorable practice towards the prevention of COVID-19. 

Question #5. If you did the associations for knowledge, attitude and practice of visitors towards COVID- 19 prevention, please rewrite your title as knowledge, attitude and practice of hospital visitors towards COVID-19 prevention....

Response: We have accepted your comment and the title was modified accordingly. 

Question #6. Bivariate analysis refers two paired data sets/with two outcome variables. But your study has single outcome so, better to use bivariable.

Response: The word bivariate analysis was replaced by bivariable analysis throughout the revised version of the manuscript.

Question #7. Where does your pretest conducted and what are the psychomotor properties of that pretest? Particularly state clearly in your methods section, than the abstract.

Response: the pretest was conducted in other districts hospital visitors (andabet hospital) other than the study area. The psychomotor domains of the pretest are reliability and validity. 

Question #8. How the response rate becomes 95.7%, if you use interviewer administered questionnaire?

Response: As we know the response rate of interviewer administered questionnaire is higher than self-administered once. But in the case of our study, the study subjects were visitors of health care facilities who came for different activities including medical treatments. The non-response rate of the study participants means that those study participants who are unable to provide data due to shortage of time, unwillingness to be part of the study and etc. 

Question #9. Most paragraphs emphasized what COVID -19 entails and may be considered as too lengthy. Introduction should follow a structured and sequential order while capturing relevant information to be included in this section. What are the current issues about COVID -19 prevention? Some of these points were included in one of your paragraphs. What are the quantifiable effect/issue around the points identified on COVID- 19 prevention? Based on the earlier identified issues about COVID-19 prevention, what are the points to be addressed in your study? What is the rationale for the study? and lastly what is the aim of the study?

Response: We thank you for this key comment. Based on your comment we tried to shorten the introduction part. Furthermore we point out the current issues of COVID-19 prevention measures, earlier identified COVID-19 prevention measures, points to be addressed, rational of the study and finally aims of the study are explained in the revised version of the manuscript (See the updated manuscript). 

Question #10. If you exclude individuals whose age is <18 years , it is better to modify your title as ... among adult visitors... otherwise, why you exclude these groups?

Response: Based on your comment we modified the title as to only adult visitors.

Question #11. You stated the total population of South Gondar Zone. However, it is better to state the average number of monthly visitors to hospitals within the zone because you are aiming to study visitors...

Response: we tried to modify this point method section particularly in study design, setting, and period of the revised version of the manuscript. 

Question #12. If these are your outcome variables, your title should be revised as KAP (knowledge, attitude, and practice of visitors towards COVID-19 prevention...) otherwise use practice as your outcome variable and use knowledge, attitude, socio-demographic.... as your independent variables. Doing research without knowing the variables is meaningless.

Response: The title of the manuscript are modified in to KAP of adult visitors towards prevention of COVID- 19 to make in line with the outcome variable of favorable/ unfavorable knowledge, attitude, and practice towards prevention of COVID-19.

Question #13. Sampling procedure is not clear

Response: Sorry for the confusion. We rewrite the sampling procedure in a more clear manner. After selecting the two hospitals randomly out of the 8 hospitals, we proportionally allocated sample size based on total estimated visitors of hospitals in the last three months. Then, 303 sample size was allocated for Debre Tabor general hospitals and 117 for Mekane Eyesus hospitals. Then hospitals visitors flow data during the previous 3 months in emergency ward, surgical ward, medical ward, gynecology/obstetrics ward and pediatrics ward considered for sample size allocation for each hospitals departments. Finally, randomly selection of visitors for each ward was selected until the allocated sample size achieved. 

Question #14. Attitude measurement is not clear. What does 26.4 (80%) score mean? is that the mean or median score of the overall attitude score?

Response: The attitude of the participants was measured using 11 items based three measurement scales with agree (3 points), neutral (2 points), and disagree (1 point). As a result the score varied from 11 to 33. Therefore, respondents with a mean score of ≥27 (80%) were considered as having a favorable attitude towards the prevention of COVID-19. 

Question #15. Conducting of pre-test and keeping of the recommended physical distances should be considered as parts of ethical consideration rather than data collection.

Response: Based on the comment we moved the statement of keeping recommended physical distance during data collection to ethical consideration from method section (Please see the ethical consideration section). 

Question #16. Is it feasible to conduct double data entery?

Response: We have written in a different term what we did, which is wrong. Thank you for your commitment in brining such errors for correction. We mean that and what we did was data entry was re-checked for 10% of the sample size in order to control data entry errors of the entered data and data cleaning was carried before statistical analysis. Thank you so much. 

Question#17. The overall knowledge of the respondents should also be stated in figures, percentages using 95% CI based on the operational definitions stated in the methods section.

Response: Based on the operational definition, the knowledge of the respondents’ was presented using figures and percentages with 95%CI. Therefore, about 280 (69.3%) (CI; 65.1-73.8) of the participants had a good knowledge towards prevention of COVID-19. 

Question #18. Your sample size is 422. However, you collect from 404 participants alone. Why? None response rate is rare in interviewer administered/ face to face questionnaire. Why this discrepancy arises?

Response: The response rate of the study was 95.7%. The response rate of interviewer administered questionnaire is higher than self-administered once. But in the case of our study, the study subjects were visitors of health care facilities who came for different activities including medical treatments. The non-response rate of the study participants means that those study participants who are unable to provide data due to shortage of time, unwillingness to be part of the study and etc. Since the study subjects were all visitors of the healthcare facilities who came for different activities including medical treatment. Therefore they withdraw from the interview for different personal reasons which made the non-response rate higher than the expected one. Of course the 95.% response rate is good for such type of study. 

Question #19. The heading of Status of Knowledge, attitude, and preventive practice of COVID-19. It is better to avoid this title and write in their own parts as I comments above.

Response: Based on your comment we omitted this subheading and the contents were written in their own parts accordingly.

Question #20. Predictor is used for more advanced studies like cohort study. In this cross - sectional study, it is better to say associated factors.

Response: Yes, this is excellent idea too. The word predictor is replaced by associated factors throughout the revised version of the manuscript.

Question #21. Discuss only your pertinent findings like knowledge, attitude, and practice rather than discussing on sources of information towards COVID-19.

Response: Thank you very much for your comment. We tried to discuss only the pertinent finding of knowledge, attitude, and practices towards prevention of COVID-19. As a result, other less important points like sources of information are removed in the revised version of the manuscript. 

Question #22. Why your justification becomes similar throughout your discussion? Please give reasonable justifications for each

Response: We tried to write specific justification for each variation in the finding of this study with other study finding 

Question #23. Conclude based on your objective. Please also include the major factors affecting practice in the conclusion section. Finally, your recommendations should be based on your results. Does it mean, there is no problem on knowledge and attitude? Please rewrite it

Response: In the original version of the manuscript our emphasis was only on prevention practice of COVID-19 rather than knowledge and attitude of visitors towards COVID-19 prevention measures. But now the title is modified in to knowledge, attitude, and practice. Therefore the conclusion is amended according to finding of the study. (See the revised version of the manuscript. 

Question #24. Avoid variables which contain a confidence interval of 1 in binary logistic regression. 

Response: All variables in logistic analysis which contains a confidence interval of 1 are excluded 

 : Question #25. Check the figure digits

Response: We have accepted the comment and all figure digits are presented with two digit value 

I hope that the revised manuscript is accepted for publication in PLoS ONE. 

Sincerely yours,

Gete Berihun 

Department of Environmental Health 

Wollo University

Dessie, Ethiopia.

---

## [Decision Letter · Decision Letter 1]

1 Apr 2021

Knowledge, Attitude, and Practices towards COVID-19 and associated factors among adult hospital Visitors in South Gondar Zone Hospitals, Northwest Ethiopia

PONE-D-20-36453R1

Dear Dr. Berihun,

We’re pleased to inform you that your manuscript has been judged scientifically suitable for publication and will be formally accepted for publication once it meets all outstanding technical requirements.

Kind regards,

Francesco Di Gennaro

Academic Editor

PLOS ONE

Additional Editor Comments (optional):

congratulations

Reviewers' comments:

Reviewer's Responses to Questions

**Comments to the Author**

1. If the authors have adequately addressed your comments raised in a previous round of review and you feel that this manuscript is now acceptable for publication, you may indicate that here to bypass the “Comments to the Author” section, enter your conflict of interest statement in the “Confidential to Editor” section, and submit your "Accept" recommendation.

Reviewer #2: (No Response)

Reviewer #3: All comments have been addressed

2. Is the manuscript technically sound, and do the data support the conclusions?

Reviewer #2: Yes

Reviewer #3: Yes

3. Has the statistical analysis been performed appropriately and rigorously? 

Reviewer #2: Yes

Reviewer #3: Yes

4. Have the authors made all data underlying the findings in their manuscript fully available?

Reviewer #2: Yes

Reviewer #3: Yes

5. Is the manuscript presented in an intelligible fashion and written in standard English?

Reviewer #2: Yes

Reviewer #3: Yes

6. Review Comments to the Author

Reviewer #2: First, I would like to congratulate the authors for exploring such an important topic. however, I have some comments and questions for them, which are found in the attached file.

Reviewer #3: fit the scientific research, all of the comment is incorporated the articles, so it published without any additional comment,

7. PLOS authors have the option to publish the peer review history of their article (what does this mean?). If published, this will include your full peer review and any attached files.

Reviewer #2: No

Reviewer #3: No

---

## [Editor Report · Acceptance letter]

6 May 2021

PONE-D-20-36453R1 

Knowledge, Attitude, and Preventive Practices towards COVID-19 and associated factors among adult hospital Visitors in South Gondar Zone Hospitals, Northwest Ethiopia 

Dear Dr. Berihun:

I'm pleased to inform you that your manuscript has been deemed suitable for publication in PLOS ONE. Congratulations! Your manuscript is now with our production department. 

Kind regards, 

on behalf of

Dr. Francesco Di Gennaro 

Academic Editor

PLOS ONE